# Why Did This Model Forecast This Future? Information-Theoretic Saliency for Counterfactual Explanations of Probabilistic Regression Models

**Chirag Raman    Alec Nonnemaker    Amelia Villegas-Morcillo    Hayley Hung    Marco Loog**

Delft University of Technology, Delft, The Netherlands
{c.a.raman, a.o.villegasmorcillo, h.hung, m.loog}@tudelft.nl
a.m.nonnemaker@student.tudelft.nl

## Abstract

We propose a post hoc saliency-based explanation framework for counterfactual reasoning in probabilistic multivariate time-series forecasting (regression) settings. Building upon Miller's framework of explanations derived from research in multiple social science disciplines, we establish a conceptual link between counterfactual reasoning and saliency-based explanation techniques. To address the lack of a principled notion of saliency, we leverage a unifying definition of information-theoretic saliency grounded in preattentive human visual cognition and extend it to forecasting settings. Specifically, we obtain a closed-form expression for commonly used density functions to identify which observed timesteps appear salient to an underlying model in making its probabilistic forecasts. We empirically validate our framework in a principled manner using synthetic data to establish ground-truth saliency that is unavailable for real-world data. Finally, using real-world data and forecasting models, we demonstrate how our framework can assist domain experts in forming new data-driven hypotheses about the causal relationships between features in the wild.

## 1 Introduction

As we go about our daily lives, engaging in conversations, walking down the street, or driving a car, we rely on our ability to anticipate the future actions and states of those around us [1, 2]. However, the numerous unknowns, such as hidden thoughts and intentions, make our predictions of the future inherently uncertain [2]. To reflect this uncertainty, several machine learning methods in such settings forecast a full distribution over plausible futures, rather than making a single point prediction [3, 4]. Identifying the factors that influence such a model's forecasts is particularly useful for domain experts seeking to understand the causal relationships guiding complex real-world behaviors, especially in situations where the future is uncertain. In this work, we introduce and address a novel research question toward counterfactual reasoning in multivariate probabilistic regression settings: how can we identify the observed timesteps that are salient for a model's probabilistic forecasts over a specific future window? Specifically, we introduce the first post hoc, model-agnostic, saliency-based explanation framework for *probabilistic* time-series forecasting.

We begin with a fundamental observation about human social cognition: we are averse to uncertainty and strive to minimize it [2]. Consider the scenario where a pedestrian is approaching you on the street. Initially, there is uncertainty about which direction each of you will take to avoid a collision. As one of you changes direction, the other observes and takes the opposite direction, ultimately avoiding a collision. Concretely, the thesis of this work is to formalize the following notion of saliency: the timestep that changes the uncertainty of a predicted future is salient toward predicting that future.

37th Conference on Neural Information Processing Systems (NeurIPS 2023).

For instance, in the aforementioned scenario, we posit that the moment when one pedestrian changes direction is salient toward forecasting the future trajectories of the pedestrians.

Our notion of saliency is grounded in preattentive human cognition and related to the concept of surprise or information associated with observations [5, 6]. Preattentive saliency captures what the brain subconsciously finds informative before conscious, or attentive, processing occurs. An unexpected or surprising observation is considered salient in this context. However, when applied to forecasting, the idea of surprisal or informativeness must be *linked to the future outcome*. Consequently, we propose that a timestep that alters an observer's certainty about the future is surprising, and therefore, salient. Crucially, our unifying 'bottom-up' perspective treats a forecasting model like a human observer, providing a principled definition of saliency that is not arbitrarily tied to task-specific error metrics. In contrast, the 'top-down' or task-specific notions of saliency common in post hoc explainable artificial intelligence (XAI) literature suffer from several drawbacks. Computed saliency maps may not measure the intended saliency, and even be independent of both the model and data generating process [7–9]. Moreover, what constitutes a *good* explanation is subject to the biases, intuition, or the visual assessment of the human observer [7, 10]; a phenomenon we refer to as the *interpretation being in the eye of the beholder*. Finally, as Barredo Arrieta et al. [11, Sec. 5.3] note, "there is absolutely no consistency behind what is known as saliency maps, salient masks, heatmaps, neuron activations, attribution, and other approaches alike."

To the best of our knowledge, no existing work addresses the specific task of obtaining post hoc model-agnostic explanations for probabilistic forecasts. Existing XAI methods for time-series data have predominantly focused on sequence classification, as we discuss in Section 2 and Appendix A. For regression, instead of post hoc explainability, researchers have emphasized interpretability by design [11] or intrinsic interpretability [12], where interpretability stems from the simple structure of models or coefficients of predefined basis functions [13, 14]. Against this backdrop, we present the following key contributions:

- Conceptual Grounding: We establish the conceptual foundation for linking saliency-based explanations with counterfactual reasoning. We draw upon insights from Miller's [10] work on explanations in artificial intelligence, highlighting the contrastive nature of explanations (Section 3).

- Information-Theoretic Framework: We extend Loog's [5] framework of bottom-up preattentive saliency to the domain of probabilistic forecasting. Specifically, we introduce a novel expression of saliency based on the differential entropy of the predicted future distribution, providing a closed-form solution for commonly used density functions in the literature (Section 4).

- Empirical Validation: We empirically evaluate our framework using synthetic and real-world data. In the synthetic setting, we achieve full accuracy in retrieving salient timesteps with known ground truth saliency. In real-world scenarios without ground truth saliency, we demonstrate the utility of our framework in explaining forecasts of social nonverbal behavior and vehicle trajectories, showcasing its effectiveness in complex and dynamic contexts (Section 5).

## 2   Related Work

**XAI Techniques for Time-Series Data.**   The taxonomy commonly used for explainability methods categorizes techniques based on three criteria: (i) intrinsic or post hoc, (ii) model-specific or model-agnostic, and (iii) local or global [12]. In the context of time-series regression, existing techniques predominantly focus on non-probabilistic settings and fall into the category of intrinsic and model-specific approaches. These include: (i) incorporating inductive biases through internal basis functions [14] (also extended to the probabilistic setting [13]), (ii) utilizing self-attention mechanisms in the model [15], and (iii) adapting saliency maps from computer vision to measure the contribution of features to the final forecast [16, 17]. For a comprehensive review of XAI methods across domains and time-series tasks, please refer to Appendix A.

**Saliency-Based Explanations and Drawbacks.**   Saliency maps gained popularity as post hoc explanation tools for image classification [16, 18]. However, the lack of consistency in defining saliency has led to diverse interpretations, including occlusion sensitivity, gradient-based attribution heatmaps, and neuron activations [11, 12]. Nevertheless, these maps are typically computed by perturbing different parts of the input and observing the resulting change in the prediction error or output class. Several issues arise with the current use of saliency maps as explanations: (i) the

feature-level manipulations used for saliency maps may distort the sample in ways that deviate from the real-world data manifold and destroy semantics [7–9]; (ii) given the arbitrary definitions, evaluating saliency maps becomes challenging and is subject to observer biases [12, Sec.10.3.2], which can lead to maps appearing correct even when they are insensitive to the model and data [7]; (iii) for forecasting, Pan et al.'s [17] notion of saliency based on the error between the point prediction and ground truth future is arbitrary and relies on ground truths unavialable during testing; and (iv) the saliency map is explicitly retrained for a single observed-future sequence, failing to capture salient patterns across similar observed sequences that result in divergent but plausible futures [17].

**Model-Agnostic Techniques.** The SHAP framework, which integrates ideas from Shapley Values, LIME, LRP, and DeepLIFT, has gained popularity as a model-agnostic approach [19]. However, adapting these techniques to time-series tasks poses several challenges. Firstly, the Shapley methods rely on functions with real-valued codomains, such as a regression function $f_x$ [19, see Eq. 4, 8], while our focus is on probabilistic models that output the distribution $p_{Y|X}$ instead of some $y = f_x(\cdot)$ to handle future uncertainty. Adapting these methods to deal with full predicted distributions is nontrivial. Similarly, gradient-based approaches compute gradients with respect to a single output instead of a full distribution. Secondly, these methods provide feature importance measures for a single output, whereas in time-series analysis, we are interested in identifying the importance of an observed timestep for an *entire future sequence*. That is, the joint consideration of the entire future sequence when computing input importance measures is challenging. As Pan et al. [17] note, in evaluating single-time predictions, these methods "ignore crucial temporal information and are insufficient for forecasting interpretation". Finally, similar to perturbation-based saliency methods, the sampling of features from arbitrary background samples in methods like Shapley/SHAP can lead to *Frankenstein Monster instances* [12, Sec. 9.5.3.3] that may not be valid samples on the data manifold. This undermines the semantics of the data, particularly in scenarios like motion trajectories, where randomly replacing features can result in physically impossible or glitchy motions.

## 3 Conceptual Grounding: Linking Saliency-Based Explanations to Counterfactual Reasoning

Given the challenges in XAI where speculations are often presented in the guise of explanations [20], we argue for grounding the concept of explanation within established frameworks of how humans define, generate, and present explanations. Turning to research in philosophy, psychology, and cognitive science, Miller [10] emphasized the importance of causality in explanatory questions. Drawing upon Pearl and Mackenzie's *Ladder of Causation* [21], he proposed the following categorization:

- Associative (*What?*): Reason about which unobserved events could have occurred given the observed events.
- Interventionist (*How?*): Simulate a change in the situation to see if the event still happens.
- Counterfactual (*Why?*): Simulate alternative causes to see whether the event still happens.

To apply Miller's framework in the context of forecasting, one needs to define the abstract notions of 'events' and 'causes'. Consider a model $\mathbf{M}$ that predicts features over a future window $\boldsymbol{t}_{\mathrm{fut}}$ by observing features over a window $\boldsymbol{t}_{\mathrm{obs}}$. We assert that the intrinsic interpretability methods involving inductive biases [13, 14] and attention mechanisms [15], fall under associative reasoning. These methods assess the (unobserved) importance of features over $\boldsymbol{t}_{\mathrm{obs}}$ using model parameters or attention coefficients based on a single prediction from $\mathbf{M}$ (the 'event') for a fixed $\boldsymbol{t}_{\mathrm{fut}}$ and single $\boldsymbol{t}_{\mathrm{obs}}$. In contrast, we posit that the perturbation-based saliency methods can support counterfactual reasoning. They perturb different parts of the input over $\boldsymbol{t}_{\mathrm{obs}}$ simulating alternative 'causes' from $\mathbf{M}$'s perspective, and observe the effect on an error metric (the 'event'). However, the current application of these methods encounters issues outlined in Section 2.

To address the aforementioned challenges, we employ a unifying information-theoretic concept of bottom-up saliency grounded in preattentive human cognition [5, 6] as discussed in Section 1. Concretely, we propose the following implication that links this saliency to counterfactual reasoning:

$$\text{observing the features at a timestep } t \in \boldsymbol{t}_{\mathrm{obs}} \text{ results in a change in } \mathbf{M}\text{'s information about the future } I_{\mathrm{fut}} \text{ over the given } \boldsymbol{t}_{\mathrm{fut}} \implies t \text{ is salient.} \tag{1}$$

Note that the antecedent (on the left of the implication) is a counterfactual statement. We formally express the implication using causal graphs [22] in Figure 1. The generic graph expresses relationships between the random variables prior to training the forecasting model $M$. The exogenous variable $\epsilon_M$ captures the randomness in the training process and modeling choices, including the distribution family for representing the forecasts. The exogenous variable $\epsilon_H$ captures the randomness in the human observer's choice of observed and future windows to examine the model. Our central idea is to evaluate the information in the model's

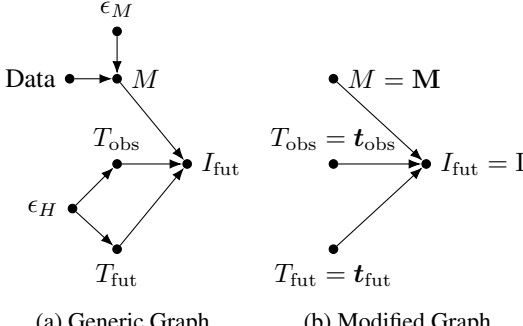

(a) Generic Graph  (b) Modified Graph

Figure 1: Causal Graphs for Explaining Forecasts

predicted distribution denoted by $I_{\text{fut}}$. Specifically, we propose posing the following counterfactual question: *What information would* $\mathbf{M}$ *have about the future over a fixed* $\boldsymbol{t}_{\text{fut}}$ *if it observed the features over* $\boldsymbol{t}_{\text{obs}}$*?* The modified graph for evaluating this question is in Figure 1b. Once the model $\mathbf{M}$ has been trained and the windows $\boldsymbol{t}_{\text{obs}}$ and $\boldsymbol{t}_{\text{fut}}$ have been chosen, the effect of the exogenous variables on the variable $I_{\text{fut}}$ disappears. This allows us to evaluate the change in the information about the future in response to *different* realizations of $\boldsymbol{t}_{\text{obs}}$ and $\boldsymbol{t}_{\text{fut}}$, facilitating counterfactual analysis. Note that we assume the modified graph is already available, as our focus is on the explanation phase. While the procedure starting from training the model in the generic graph implicitly follows Pearl's *abduct-action-prediction* process [22, p. 207], estimating the distribution over the exogenous variables from the *abduction* step is conceptually not applicable in this setting.

Note that these graphs are not meant to describe relationships between random variables in the data for a specific hypothesis, as is typical in causal inference literature: for instance, the effect of [rotating toward the speaker] on [turn changes] in conversations. Rather, they describe the *process of a human generating contrastive explanations* for a given pretrained forecasting model $M$—irrespective of whether or not it is the true model—for some sequences in the data $(\boldsymbol{t}_{\text{obs}}, \boldsymbol{t}_{\text{fut}})$. Further, the notion of counterfactuals, as used within the context of contrastive explanations, is also distinct from that in causal inference. As Miller [10, Sec. 2.3] points out, "it is important to note that this is not the same counterfactual that one refers to when determining causality. For causality, the counterfactuals are hypothetical 'noncauses'... whereas in contrastive explanation, counterfactuals are hypothetical outcomes." Miller's point is that *why* explanations entail contrastive reasoning which involves comparing 'outcomes' in response to alternate 'causes'. In our work, this 'outcome' relates to the information in $M$'s predicted distribution, the what-if question being "would the information in $M$'s prediction for the window $\boldsymbol{t}_{\text{fut}}$ change if it had observed features over a different (contrastive) $\boldsymbol{t}_{\text{obs}}$?". Contrast this to associative reasoning which uses features from a single $\boldsymbol{t}_{\text{obs}}$ to generate the attribution map. A longer discussion is in Appendix E.

Considering the information in forecasts in implication 1 links counterfactual reasoning to a more principled notion of saliency than has been used in XAI literature. Note that the implication entails that for the antecedent to be true $t$ must be salient. However, knowing the antecedent is false is not sufficient to conclude that $t$ is not salient, i.e. there can be other notions of saliency that make $t$ salient. However, for less speculative evaluation, it is crucial that we use a unifying notion of saliency that is not arbitrarily defined based on task-specific error metrics or model gradients [11, 16–18]. Preattentive saliency, as we formalize in Section 4.1, is based on what is informative for the brain *before* conscious processing, making it more objective in nature.

Our framework addresses all the concerns associated with saliency-based approaches described in Section 2: (i) the counterfactuals in our framework are real observed features rather than random input perturbations, preserving the semantics of the real-world data; (ii) our use of information-theoretic preattentive saliency is principled and objective; (iii) our framework allows for saliency computation on unseen test data where the ground-truth future is unavailable, relying solely on the underlying model; and (iv) our approach considers the distribution over possible futures for a single input, capturing the structural predictive relationships between features across multiple samples. An additional advantage of our framework is that it does not require any training to compute the saliency and can be applied to any model that outputs a distribution over futures.

# 4 Methodology: Closed-Form Saliency for Probabilistic Forecasting

## 4.1 Preliminary: Information Theoretic Preattentive Saliency

Loog [5] developed a general closed-form expression for saliency based on computational visual perception that unifies different definitions of saliency encountered in the literature. The framework was illustrated on images and employed a surprisal-based operational definition of bottom-up attention. In this framework, an image is represented by a feature mapping function $\phi$ that relates each location in the image to a set of features. The saliency of a location $x$ is determined by the information or surprise associated with its corresponding feature vector $\phi(x)$ compared to other feature vectors extracted from the same image. The saliency measure is defined as follows:

$$S(x) > S(x') \iff -\log p_\Phi(\phi(x)) > -\log p_\Phi(\phi(x')). \tag{2}$$

Here, $p_\Phi$ represents the probability density function over all feature vectors, while $p_X$ captures any prior knowledge that influences the saliency of different image locations.

Contrary to approaches that determine saliency maps through an explicit data-driven density estimation [16, 18, 23–25], once the feature mapping $\phi$ is fixed, a closed-form expression for saliency can be obtained. The information content $-\log p_\Phi$ can be obtained from $\log p_X$ through a simple change of variables [26] from $x$ to $\phi(x)$. The saliency $S(x)$ is then given by the expression:

$$-\log p_\Phi(\phi(x)) = -\log p_X(x) + \frac{1}{2}\log\det(J_\phi^t(x)J_\phi(x)), \tag{3}$$

where $J_\phi$ denotes the Jacobian matrix of $\phi$, and $\_^t$ indicates matrix transposition. Since a monotonic transformation does not essentially alter the map, Loog [5] simplifies the saliency map definition to

$$S(x) := \det(J_\phi^t(x)J_\phi(x)), \tag{4}$$

This formulation of saliency offers several advantages. It provides a principled and objective measure that captures the informativeness of features for human perception. Moreover, the saliency computation is purely local to an image, making it independent of previously observed data.

## 4.2 Defining $\phi$ in Terms of the Uncertainty over the Future Window $t_{\text{fut}}$

Let $t_{\text{obs}} := [o1, o2, ..., oT]$ represent a window of consecutively increasing observed timesteps, and $t_{\text{fut}} := [f1, f2, ..., fT]$ denote an unobserved future time window, where $f1 > oT$. Consider a set of $n$ interacting agents, and let $X := [b_t^i; t \in t_{\text{obs}}]_{i=1}^n$ and $Y := [b_t^i; t \in t_{\text{fut}}]_{i=1}^n$ represent their features over $t_{\text{obs}}$ and $t_{\text{fut}}$ respectively. Here, $b_t^i$ captures multimodal features from agent $i$ at time $t$. The forecasting task is to predict the density $p_{Y|X}$. Given a model that outputs $p_{Y|X}$, our task is to compute the saliency $S(t_{\text{obs}})$ of an observed $t_{\text{obs}}$ with respect to a fixed choice of $t_{\text{fut}}$.

To extend Loog's [5] framework to forecasting settings, we need to choose an appropriate $\phi$. We formalize the implication in Equation 1 and map $t_{\text{obs}}$ to the differential entropy of the model's predicted future distribution over $t_{\text{fut}}$. Specifically, we define $\phi : t_{\text{obs}} \mapsto h(Y|X = X)$, where the conditional differential entropy of $Y$ given $\{X = X\}$ is defined as

$$h(Y|X = X) := -\int p_{Y|X}(Y|X)\log p_{Y|X}(Y|X)dY. \tag{5}$$

Our framework is summarized in Algorithm 1. Consider that a domain expert selects a specific $t_{\text{fut}}$ corresponding to a high-order semantic behavior they wish to analyze. This could be a speaking-turn change [27, 28] an interaction termination [29, 30], or a synchronous behavior event [31]. Given an underlying forecasting model $M$ and look-back period before $t_{\text{fut}}$, we compute $h(Y|X = X)$ for different *observed multivariate features* $X$ corresponding to different locations of a sliding $t_{\text{obs}}$. The computed differential entropy values are then inserted into Equation 4 to obtain the saliency of different $t_{\text{obs}}$ locations towards the future over the chosen $t_{\text{fut}}$. In Appendix B we discuss other favorable properties of differential entropy that make it a suitable choice as $\phi$.

**Explanation Using the Running Example.** Within our running example from Section 1, $t_{\text{fut}}$ corresponds to the two pedestrians passing each other while avoiding collision. In this example, let us assume $M$'s training data contains examples of pedestrians passing others to both the left and the

---

**Algorithm 1** Temporal Saliency in Probabilistic Forecasting

---

**Input:** The probability density function $p_{Y|X}$, a fixed $t_{\text{fut}}$ of interest, a sequence of $m$ preceding observed windows $O = [t_{\text{obs}}^1, \ldots, t_{\text{obs}}^m]$, and the behavioral features $X^j$ for every $t_{\text{obs}}^j$

**Output:** The saliency map $S(O)$ over the observed windows

  1: **for each** $t_{\text{obs}}^j \in O$ **do**

  2:    Compute the feature mapping $\phi(t_{\text{obs}}^j) \leftarrow h(Y|X = X^j)$

  3: **end for**

  4: Compute saliency $S(t_{\text{obs}}) \leftarrow \det(J_\phi^t(t_{\text{obs}}) J_\phi(t_{\text{obs}}))$

---

right. Consequently, for a $t_{\text{obs}}$ containing the pedestrians approaching each other in a straight line, the predicted distribution $p_{Y|X}$ over $t_{\text{fut}}$ encapsulates both possibilities of each pedestrian passing to the left as well as the right of the other. So the entropy $h(Y|X = X)$ is high for this $t_{\text{obs}}$. Only once $M$ is fed as input with the trajectories from the $t_{\text{obs}}$ containing the pedestrians choosing one of the two directions to pass, the predicted $p_{Y|X}$ is certain in terms of the pedestrians continuing along the chosen direction. (Note that in this case, we assume $M$ has been trained by maximizing likelihood over the dataset containing only these two direction changes for avoiding collision.) Consequently, the entropy $h(Y|X = X)$ drops only once this moment of the pedestrians committing to a direction is seen by the model and would be considered salient for our algorithm.

### 4.3 Computing $h(Y|X = X)$

Typically, the density $p_{Y|X}$ is modeled as a multivariate Gaussian distribution [4, 32–34]. When the decoding of the future is non-autoregressive, the parameters of the distributions for all $t \in t_{\text{fut}}$ are estimated at once, and the differential entropy has a closed-form expression, given by (see Cover and Thomas [35, Theorem 8.4.1])

$$h(Y|X = X) = h(\mathcal{N}_d(\boldsymbol{\mu}, \boldsymbol{K})) = \frac{1}{2} \log[(2\pi e)^d \det(\boldsymbol{K})]. \tag{6}$$

A common choice is to set $\boldsymbol{K}$ to be diagonal, i.e. the predicted distribution is factorized over agents and features. In this case, we can simply sum the $\log$ of the individual variances to obtain the feature mapping $\phi$. Note that from Equation 6, for a multivariate Gaussian distribution, the differential entropy only depends on the covariance, or the *spread* of the distribution, aligning with the notion of differential entropy as a measure of total uncertainty. (See [35, Tab. 17.1; 36] for closed-form expressions for a large number of commonly employed probability density functions.)

In cases where probabilistic autoregressive decoders are used [4, 33, 37, 38], we do not have access to the full joint distribution $p_{Y_{f1}, \ldots, Y_{fT}|X}$ for the timesteps in $t_{\text{fut}}$. This is because inferring the density function $p_{Y|X}$ often involves sampling: a specific sample $\widehat{Y}_t$ is taken from the predicted density at each $t \in t_{\text{fut}}$, and passed back as input to the decoder for estimating the density at timestep $t + 1$ [37, 38]. Therefore, the density at $t + 1$ depends on the randomness introduced in sampling $\widehat{Y}_t$. Figure 2 illustrates the concept for two timesteps. Here, a single forecast would only output the shaded red distribution for $Y_2$. In such cases, computing the joint entropy $h(Y_1, Y_2)$ directly is challenging in the absence of the full joint distribution $p_{Y_1, Y_2}$.

To address this, we have two options. The simpler option is to redefine our feature-mapping as $\phi : t_{\text{obs}} \mapsto \sum_{t \in t_{\text{fut}}} h(Y_t|\widehat{Y}_{<t}, X)$, i.e. we approximate the total uncertainty over the predicted sequence by summing the differential entropies of the individual densities estimated at

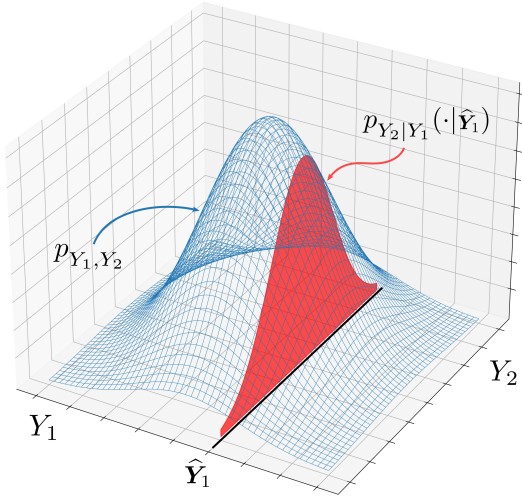

Figure 2: Illustrating predicted densities under greedy autoregressive decoding for two timesteps. For simplicity, we depict a joint Gaussian distribution and omit the conditioning on $X$ everywhere.

each timestep. Note that following the chain rule for differential entropy (see Cover and Thomas [35, Eq. 8.62]), the joint entropy can indeed be written as the sum of individual conditionals. However,

$$h(Y|X = \boldsymbol{X}) = \sum_{t \in \boldsymbol{t}_{\text{fut}}} h(Y_t|Y_{<t}, \boldsymbol{X}) \neq \sum_{t \in \boldsymbol{t}_{\text{fut}}} h(Y_t|\widehat{\boldsymbol{Y}}_{<t}, \boldsymbol{X}). \tag{7}$$

And yet, training autoregressive decoders by maximizing likelihood actually assumes the inequality in Equation 7 to be approximately equal (see [39, Sec. 2; 40, Eq. 5]). The approximation relies on the observation that, for autoregressive decoding, the parameters of the predicted distribution for $Y_t$ are computed as a deterministic function of the decoder hidden state. That is, $Y_t$ is conditionally independent of $Y_{<t}$ given the hidden state of the decoder $\boldsymbol{s}_t$ at timestep $t$. The underlying assumption is that for a well-trained decoder, $\boldsymbol{s}_t$ encodes all relevant information from other timesteps to infer the distribution of $Y_t$. So at inference, despite being a function of the single sample $\widehat{\boldsymbol{Y}}_{t-1}$, the predicted distribution conditioned on $\boldsymbol{s}_t$ provides a reasonable estimate of the uncertainty in $Y_t$. This assumption allows us to again obtain a closed-form expression for the saliency map when each $Y_t$ is modeled using a density function with a known closed-form expression for differential entropy [35, Tab. 17.1; 36]. For the common choice of modeling $Y_t$ using a Gaussian mixture [37, 38], approximations that approach the true differential entropy can also be obtained efficiently [41–43] to directly compute the feature mapping $\phi$.

The second option is to estimate $h(Y|X = \boldsymbol{X})$ using sampling or other non-parametric approaches when analytical expressions or computationally efficient approximations are not available [44–47]. These sampling-based methods provide approximations that converge to the true entropy, although they may be computationally more expensive than parametric methods. Overall, the choice of modeling the future density and the approach for estimating the differential entropy depends on the specific scenario and the available resources.

## 5 Experiments

The common evaluation of saliency-based explanations relies on qualitative visual assessment, which is subjective and prone to observer biases [7, 11, 12]. Meanwhile, establishing a reliable ground truth for the salient relationship between the observed window $\boldsymbol{t}_{\text{obs}}$ and the future window $\boldsymbol{t}_{\text{fut}}$ is challenging in real-world data due to conflicting domain evidence on predictive relationships [28, 48]. Furthermore, fair validation of a *model agnostic, post hoc* method requires evaluating it independently of imperfections in the underlying forecasting model. To address these challenges we conduct two types of empirical evaluation: one using synthetic data to establish ground truth predictive saliency and *validate the framework*, and another to *demonstrate empirical utility* in real-world scenarios where perfect forecasts and ground truth saliency are unavailable.

No existing benchmarks or post hoc explanation frameworks exist for probabilistic time-series regression that meet the necessary requirements for a meaningful empirical comparison. Nevertheless, we provide results by adapting several explainability frameworks in our experiments. Specifically, we considered DeepSHAP and GradientSHAP [19], and IntegratedGradients and SmoothGrad [49]. It is important to note that we do not imply that these are fair comparisons; they are not (see Section 2). However, the comparisons are meant to characterize results from popular tools that practitioners are likely to use in the absence of our proposed framework. Implementation details for the following experiments and additional results for the real-world scenarios are in Appendices C and D, respectively.

### 5.1 Empirical Validation using Synthesized Ground Truth Saliency

**Dataset.** We simulate a group conversation that emulates real behavior patterns. Listeners typically focus on the speaker, while the speaker looks at different listeners [50]. Additionally, head gestures and gaze patterns predict the next speaker [51–54]. In our simplified simulation, the speaker rotates towards the center when speaking, and listeners nod to trigger a turn handover. We use real-valued quaternions to represent 3D head poses, commonly used for human motion and pose representation [4, 55, 56]. Following the notation in Section 4.2, $\boldsymbol{b} = [q_w, q_x, q_y, q_z, ss]$ where $ss$ denotes binary speaking status. We simulate the turn changes to occur once clockwise and once anticlockwise. The ground truth salient timestep is when a listener initiates a head nod to trigger a turn handover, ensuring a certain future turn change. Figure 3 illustrates this mechanism. The code, dataset, and animated visualization are available in the Supplement.

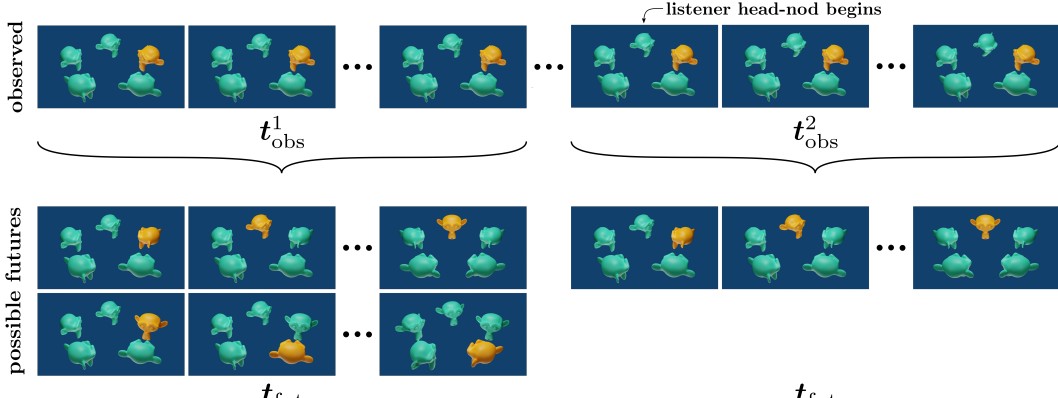

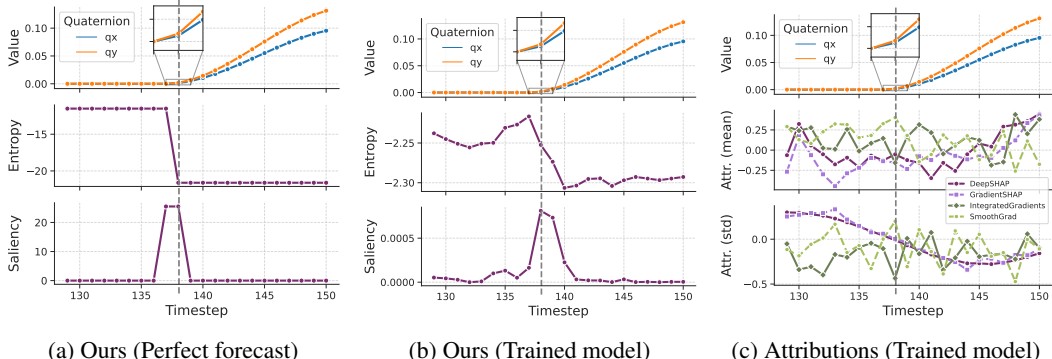

Figure 3: **Illustrating the synthetic conversation dynamics dataset.** Speakers are denoted in orange and listeners in green. For a fixed $t_{\text{fut}}$ we depict two preceding $t_{\text{obs}}$ windows. By construction, when observing a stable speaking turn over $t_{\text{obs}}^1$, two valid futures are possible over $t_{\text{fut}}$. These correspond to a turn handover to the immediate left or right of the current speaker. Over $t_{\text{obs}}^2$, when a listener nods to indicate the desire to take the floor, the future over $t_{\text{fut}}$ becomes certain, corresponding to the listener successfully taking over the speaking turn. Here $t_{\text{obs}}^2$ is consequently more salient than $t_{\text{obs}}^1$ towards forecasting the turn change over $t_{\text{fut}}$. (Best viewed as video, see Supplement.)

(a) Ours (Perfect forecast)    (b) Ours (Trained model)    (c) Attributions (Trained model)

Figure 4: **Computing Saliency**. The top plots show the quaternion dimensions *qx* and *qy* for the listener that nods over $t_{\text{obs}}^2$ in Figure 3. The gray dotted line indicates the true salient timestep 138 when the head nod begins, making the future over timesteps $183 - 228$ ($t_{\text{fut}}$) certain. The rest of plots show the **(a)** entropy over future values of all participants (middle), and saliency map obtained using our framework (bottom), considering perfect forecasts; **(b)** entropy and saliency for the forecasts from a Social Process model; and **(c)** mean attributions across features per timestep from different explainability frameworks (DeepSHAP, GradientSHAP, IntegratedGradients, and SmoothGrad) for the predicted mean and std. of the same forecast from the Social Process model.

**Empirical Validation.** To validate our framework in isolation, we assume a perfect forecasting model that predicts the true distribution over the possible future quaternion trajectories. The forecasting model focuses solely on low-level features and does not incorporate any high-order semantics of turn-taking. The saliency map generated by our framework, as shown in Figure 4a, accurately identifies the ground truth salient timesteps at frames 138 and 139 where the head nod begins. The saliency decreases once the nod is already in motion, indicating that it does not provide additional information about the future. This empirically validates our framework.

**Introducing a Real Forecasting Model.** We evaluate our framework using a real underlying forecasting model trained on synthetic data. We employ a *Social Process* model [4] for its ability to capture relative partner behavior and adapt to specific group dynamics. As shown in Figure 4b, our framework identifies the true salient timesteps with higher saliency values. Conversely, the attributions provided by other explainability frameworks in Figure 4c for the predicted mean and

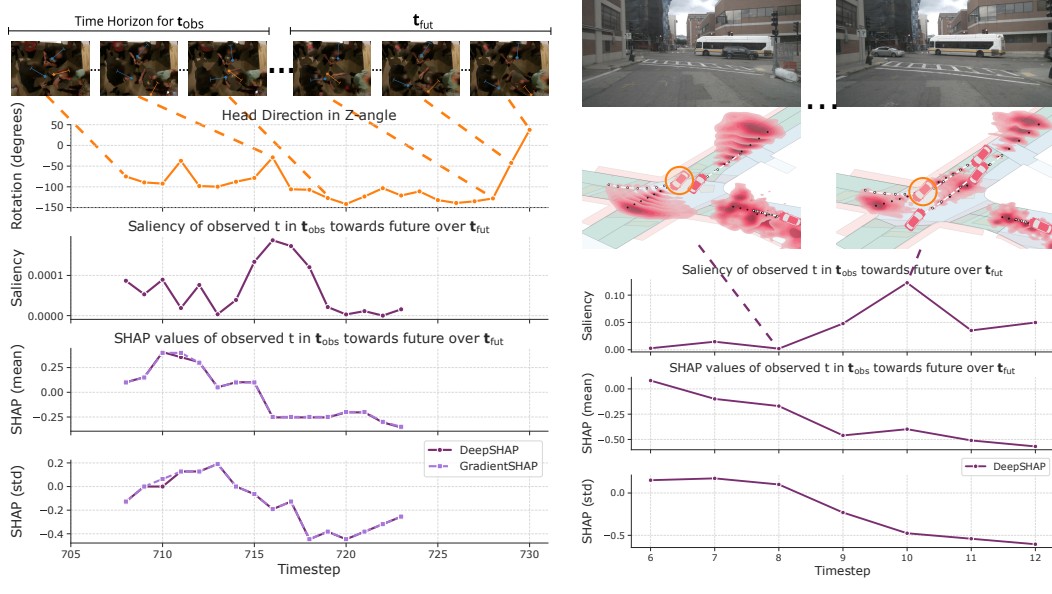

(a) Group leaving behavior (*MatchNMingle* dataset)      (b) Vehicle trajectory (*nuScenes* dataset)

Figure 5: **(a) Analysis of the group leaving instance** at 12:11 on *Day 1, Cam 12* in the *MatchNMingle* dataset. ***Row 1***: Video frames and overlaid arrows denoting the head orientation of participants. Orange indicates the person leaving the group over the $t_{\text{fut}}$. ***Row 2***: Head orientation of the leaver plotted as 2D horizontal rotation. ***Row 3***: Saliency map from running predictions from the Attentive Social Process model through our framework. The timesteps salient towards the model's forecasts correspond to the leaver making sweeping gazes away from the group. ***Rows 4-5***: mean DeepSHAP and GradientSHAP across features per timestep for the predicted mean and std. of the same forecast. **(b) Analysis of the vehicle turn making instance** on *Scene 3* in the *nuScenes* dataset. ***Row 1***: Video frames showing the bus and surrounding cars from the camera. ***Row 2***: Future predictions for the bus position (circled) from the Trajectron++ model (ground truth in white, predicted mean in black and variance in red). ***Row 3***: Saliency map from running predictions through our framework. The timesteps salient correspond to the model being more certain that the bus will make a turn. ***Rows 4-5***: mean DeepSHAP values across features per timestep for the predicted mean and std. of the same forecast. Best viewed as video (see Supplement).

standard deviation of the same forecast fail to capture the salient predictive relationships in the data. This comparison underscores the effectiveness of our framework in capturing meaningful and interpretable saliency, even in conjunction with an imperfect forecasting model.

## 5.2 Empirical Evaluation in Real-World Scenarios

### 5.2.1 Group Leaving Behavior in the Wild

The study of group-leaving behavior has garnered interest in social psychology and the development of conversational agents [29, 30]. Recent approaches employ data-driven models to predict future non-verbal cues, capturing general predictive patterns in the data [4]. In this study, we demonstrate how our framework can assist domain experts in hypothesizing about the causal relationships between behavioral patterns and group leaving. We leverage the publicly available *MatchNMingle* dataset [57], which features natural interactions of 92 individuals during a cocktail party. We use an *Attentive Social Process* model [4] to forecast continuous head pose, body pose, and binary speaking status.

Through our analysis (see Figure 5a), we find that the salient timesteps in the model's forecasts correspond to instances when a person about to leave directs their gaze away from the shared group space (*o-space* [1]) by rotating their head. This observation leads to the following hypothesis:

> gazing away from the o-space of a conversing group is predictive of group leaving.

While this hypothesis aligns with established leave-taking patterns [1, 58] and the sweeping gaze behavior associated with seeking new interaction partners [59], it requires further validation through subsequent studies and rigorous statistical testing with the involvement of domain experts. Nonetheless, our experiment demonstrates how the framework can unveil data-driven insights into patterns that, in other cases, may have been overlooked by humans but captured by the forecasting model. By contrast, we do not observe any discernible intuitive patterns in the features associated with the trends in DeepSHAP and GradientSHAP values for the predicted mean and standard deviation.

### 5.2.2 Vehicle Trajectory Forecasting

The accurate forecasting of pedestrian and vehicle trajectories is crucial for safe and socially-aware autonomous navigation of vehicles [37, 60–62]. In this study, we utilize our framework to investigate vehicle dynamics in real driving scenarios. Specifically, we leverage the *nuScenes* dataset, a multimodal dataset for autonomous driving [63], and the *Trajectron++* forecasting model [37].

Figure 5b illustrates our analysis of vehicle dynamics at an intersection. Notably, our framework identifies a salient timestep for the Trajectron++ model precisely when it becomes more confident that the bus will make a turn instead of continuing straight. This coincides with the model's increased certainty that the point-of-view vehicle will decelerate as a new vehicle enters the scene from the left. Although there are no relevant domain-specific theories in this case to interpret this saliency, these identified patterns align with expected driving behavior. In contrast, the DeepSHAP values fail to capture the model's change in certainty about the bus making the turn instead of continuing straight. Moreover, we also do not identify any intuitive patterns in the predictions associated with the DeepSHAP trends. Thus, our framework serves as a valuable tool for sanity-checking model forecasts in real-world driving scenarios. It helps identify instances where the model's predictions align or misalign with established norms and expectations.

## 6 Conclusion

We have proposed a computational framework that provides counterfactual explanations of model forecasts based on a principled notion of bottom-up task-agnostic saliency. We derive a closed-form expression to compute this saliency for commonly used probability density functions to represent forecasts [4, 37, 38, 62]. To validate our framework, we conduct empirical experiments using a synthetic setup, enabling quantitative validation and mitigating observer biases associated with visual assessment of saliency maps. Additionally, we demonstrate the practical utility of our framework in two real-world scenarios involving the prediction of nonverbal social behavior and vehicle trajectories. By identifying salient timesteps towards a predicted future through counterfactual reasoning, our framework can support domain experts in formulating data-driven hypotheses regarding the predictive causal patterns involving the *features* present at those salient timesteps. These hypotheses can then be tested through subsequent controlled experiments, establishing a human-in-the-loop Explainable AI (XAI) methodology. For a more comprehensive discussion, please refer to Appendix E.

## 7 Limitations and Potential Negative Societal Impact

While our framework provides a closed-form or efficient solution for most probability density functions, limitations arise when an analytic expression for differential entropy is unavailable. As discussed in Section 4.3, alternative approaches like sampling or nonparametric methods can be employed to approximate the entropy, albeit at an increased computational cost.

Our work here is an upstream methodological contribution. However, when applied downstream to human behavior or healthcare data, ethical considerations arise naturally. Here, care must be taken that such methods are not applied for gaining insights into behavior in a way that violates the privacy of people. Our framework enables domain experts to derive data-driven insights and hypotheses about predictive causal patterns. However, hypotheses should be rigorously tested, using controlled experiments and peer review, before being considered valid statements about human behavior. Collaboration among researchers, practitioners, and policymakers across disciplines is crucial to mitigate such societal risks and ensure ethical deployment of AI technologies.

## Acknowledgments

The authors would like to thank Jesse Krijthe, Rickard Karlsson, David Tax, Yeshwanth Napolean, and Megha Khosla for the thoughtful discussions.

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

# A   Broader Related Work: Explainable Methods for Time-Series Data Across Tasks and Domains

The larger focus of explainability techniques involving time-series data has been on the task of **classifying** time-series. The goal has been to estimate the relevance of each input feature at a given timestep towards each output class. Here, saliency approaches often overlap with techniques developed for image data and can be categorized into:

**Gradient-Based Techniques.** The broad approach involves evaluating the gradient of the output class with respect to the input [64]. Several variants have been proposed [19, 49, 65–67].

**Perturbation-Based Techniques.** The idea is to examine how the output changes in response to some perturbation of the input. Perturbations are implemented by either occluding contiguous regions of the input [68, 69]; performing an ablation of the features [70]; or randomly permuting features [12]. Ismail et al. [71] provide a benchmark of a subset of these techniques.

**Attention-Based Techniques.** These incorporate an attention mechanism into the model that is trained to attribute importance to different parts of the input sequence towards a prediction at each future timestep. Such techniques have been extensively utilized for healthcare data. Early methods applied a reverse-time attention [72], Later methods applied the attention to probabilistic state-space representations [73].

Some of these broad ideas have been applied to the **regression** setting to make interpretable forecasts of future time-series features. Lim et al. [15] leveraged self-attention layers for capturing long-term dependencies. Pan et al. [17] recently proposed computing saliency as a mixup strategy between series images and their perturbed version with a learnable mask for each sample. They view saliency in terms of minimizing the mean squared error between the predictions and ground-truths for a particular instance. Focusing on the univariate point-forecasting problem, Oreshkin et al. [14] proposed injecting inductive biases by computing the forecast as a combination of a trend and seasonality model. They argue that this decomposition makes the outputs more interpretable.

Developing explainable techniques for the probabilistic forecasting setting remains largely unexplored and subject to non-overlapping notions of explainability. Rügamer et al. [13] transform the forecast using predefined basis functions such as Bernstein polynomials. They relate interpretability to the coefficients of these basis functions (a notion similar to that of Oreshkin et al. [14]). Panja et al. [74] embed the classical linear ARIMA model into a non-linear autoregressive neural network for univariate probabilistic forecasting. As before, the explainability here also stems from the 'white-box' nature of the linear ARIMA component. Li et al. [75] propose an automatic relevance determination network to identify useful exogenous variables (i.e. variables that can affect the forecast without being a part of the time-series data). To the best of our knowledge, saliency-based methods have not yet been considered within this setting.

# B   Favorable Properties of Differential Entropy

Differential entropy possesses favorable properties that make it a suitable choice as $\phi$ for computing the saliency map. First, the scale of the forecast density does not affect the resulting saliency map (see Cover and Thomas [35, Theorem 8.6.4]):

$$h(aY) = h(Y) + \log|a|, \text{ for } a \neq 0, \text{ and} \tag{8}$$

$$h(\boldsymbol{A}Y) = h(Y) + \log|\det(\boldsymbol{A})|, \text{ when } \boldsymbol{A} \text{ is a square matrix.} \tag{9}$$

That is, scaling the distribution changes the differential entropy by only a constant factor. So the saliency map resulting from inserting the entropy into Equation 4 remains unaffected since the Jacobian term only depends on the relative change in entropy across different choices of $\boldsymbol{t}_{\text{obs}}$. Similarly, translating the predicted density leaves the saliency map unaffected (see Cover and Thomas [35, Theorem 8.6.3]):

$$h(Y + c) = h(Y). \tag{10}$$

# C    Implementation Details for Experiments

## C.1    Other Explainability Methods

For DeepSHAP and GradientSHAP, we used the official implementation of SHAP: `https://github.com/slundberg/shap`. For IntegratedGradients and SmoothGrad, we used the Captum framework: `https://captum.ai/`. We reiterate that these are not fair comparisons, for the reasons we have discussed in Section 2. One crucial reason is that no existing method is designed to handle a predicted distribution. To apply them in this context, we need to compute attribution values for the predicted mean and standard deviation for every feature at every $t \in \boldsymbol{t}_{\text{fut}}$ in isolation. In contrast, by measuring differential entropy, our method jointly accounts for the parameters of the distribution and captures the information content of the distribution. Despite these limitations, we include these comparisons to provide readers with a contextual understanding of the results obtained from commonly used explainability tools.

**Computational Efficiency.**    For further insight, we measured the execution time of our saliency method compared to DeepSHAP for both real-world scenarios (see Table 1). The main takeaway is that our method is an order of magnitude faster at computing the saliency map compared to DeepSHAP. This is in part because DeepSHAP computes values for every feature at every timestep in the output independently. Even if this is parallelized, DeepSHAP requires multiple forward passes and gradient computations including samples from a background set to compute the reference values. The computation time scales with the size of the background set.

Table 1: **Comparing Computational Efficiency.** We compare the practical execution time of our method to running DeepSHAP. For a reasonably fair characterization of DeepSHAP, we report execution time for computing the SHAP values associated with a single predicted parameter, the *mean* of the future distribution.

| Scenario | Saliency | | DeepSHAP | |
| --- | --- | --- | --- | --- |
|  | **Forward Pass** | **Compute Saliency** | **Init** | **Compute Values** |
| Group leaving (*MatchNMingle* dataset) | 50 ms | 2 ms | 50 ms | 13.5 min |
| Vehicle trajectory (*nuScenes* dataset) | 2.5 s | 33 ms | 68 ms | 26.3 min |

## C.2    Empirical Validation using Synthetic Data

We model the future distribution using a Gaussian function for simplicity (setting std. to $10^{-10}$ for the single future), but a more complex distribution that predicts the appropriate change in variance would also work in practice. We now implement Algorithm 1 as follows. We identify a window where a turn change occurs in the data (frames 183-228) and denote this 45 frame window as the $\boldsymbol{t}_{\text{fut}}$ of interest. While we manually identify an interesting event for illustration, such a window could also correspond to an interesting prediction by a model. We generate a set of candidate $\boldsymbol{t}_{\text{obs}}$ by sliding a 30 frame window over a horizon of 100 frames prior to $\boldsymbol{t}_{\text{fut}}$, with a stride of 1 frame. For every observed $\boldsymbol{t}_{\text{obs}}$, we fit a Gaussian density to the quaternion and speaking status features of all participants over the futures that can occur during $\boldsymbol{t}_{\text{fut}}$. We then set the entropy of this Gaussian density as the feature $\phi$ for that $\boldsymbol{t}_{\text{obs}}$. For the experiments with a real forecasting model, we employ the `[recurrent, uniform attention]` variant of the Social Process family given its ability to capture dynamic movements [4].

## C.3    Forecasting Group Leaving Behavior

We used the pretrained model on the *MatchNMingle* dataset provided with the official implementation of Social Processes [4]: `https://github.com/chiragraman/social-processes`. Specifically, we employed the `[recurrent, dot-attention]` variant of the Attentive Social Process family (ASP-GRU-dot). We set $\boldsymbol{t}_{\text{fut}}$ to correspond to a 3 second window (3 frames in the data, which is at 1 Hz) containing an individual leaving a conversing group. We obtained forecasts from the model corresponding to a rolling 5 second $\boldsymbol{t}_{\text{obs}}$ within a 20 second preceding horizon. For computing DeepSHAP and GradientSHAP values, we used the entire observed time horizon as the background

dataset. This ensures that the expected values are computed within a reasonably similar context for a given $t_{\text{obs}}$.

## C.4 Vehicle Trajectory Forecasting

We trained the Trajectron++ model [37] on the *nuScenes* dataset (mini version) [63] using the default command provided in the official implementation: `https://github.com/StanfordASL/Trajectron-plus-plus`. In particular, we used the `int_ee` model which incorporates the agent's system dynamics to produce dynamically-feasible trajectories. For analysis, we used a sequence from scene index 3 (`scene.name` 757). We set $t_{\text{fut}}$ to correspond to a window of 6 timesteps (14-19) with a lookback horizon from timesteps 6 to 13. For computing DeepSHAP values we again used the entire observed horizon as the background set. The features in the observed sequence contained several `NaN` entries, which resulted in `NaN` DeepSHAP values. We ignored these when aggregating values per timestep.

## D   Additional Results

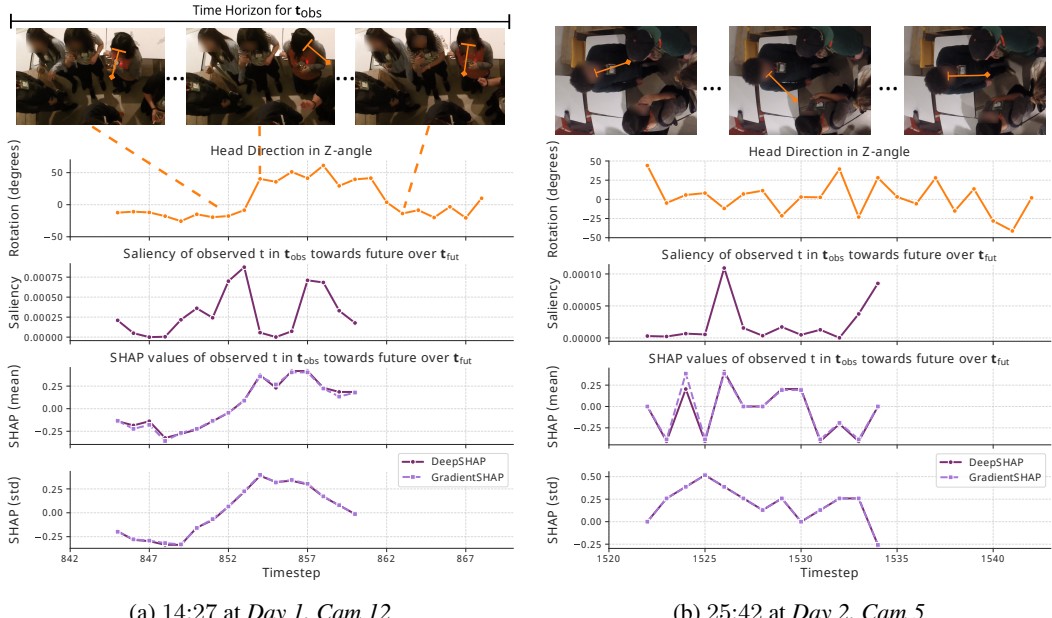

(a) 14:27 at *Day 1, Cam 12*      (b) 25:42 at *Day 2, Cam 5*

Figure 6: **Analysis of two sequences in the *MatchNMingle* dataset**. *Row 1*: Video frames and overlaid arrows denoting the head orientation of the participant of interest. *Row 2*: Head orientation plotted as 2D horizontal rotation. *Rows 3-5*: Saliency map from running predictions from the Attentive Social Process model through our framework, as well as the mean DeepSHAP and GradientSHAP values across features per timestep for the predicted mean and std. of the same forecast. Our saliency framework succeeds in identifying the salient timesteps in both cases. In **(a)**, the timestep in which the participant of interest is looking away is identified as salient towards predicting the other participant leaving the dyadic interaction. In **(b)**, the timestep in which the participant of interest suddenly stops actively participating in the conversation (not nodding or looking at the speakers) is identified as salient towards predicting their group leaving.

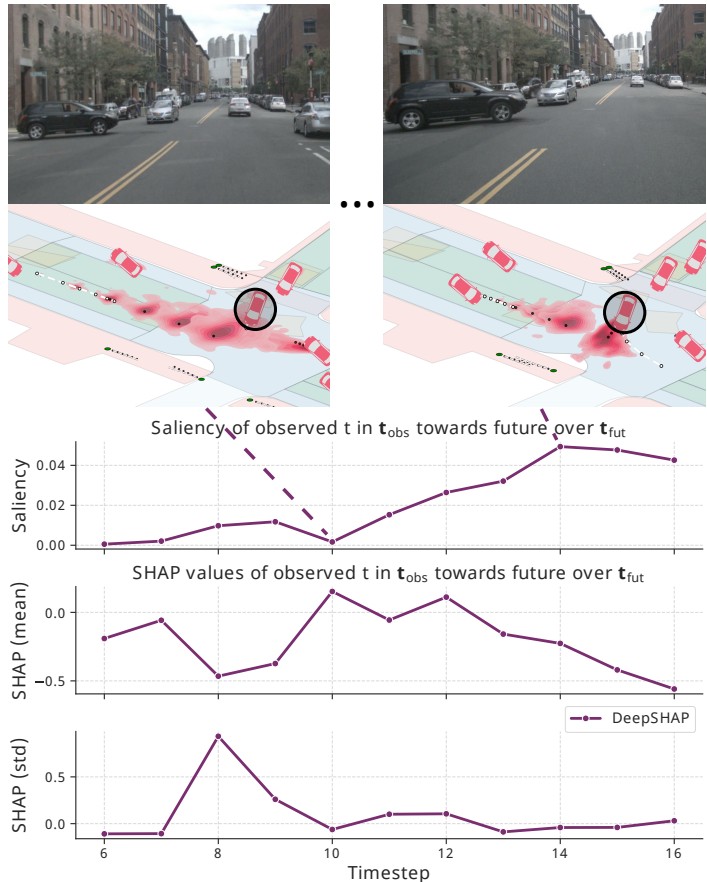

Figure 7: **Analysis of a sequence on *Scene 0* in the *nuScenes* dataset**. ***Row 1***: Video frames showing the black turning car and surrounding cars from the camera. ***Row 2***: Future predictions for the black car position (circled) from the Trajectron++ model (ground truth in white, predicted mean in black and variance in red). ***Rows 3-5***: Saliency map from running predictions through our framework, as well as the mean DeepSHAP values across features per timestep for the predicted mean and std. of the same forecast. Our framework reveals that the silver and point-of-view cars slowing is salient for the model in predicting that the black car completes the turn.

# E Broader Discussion: Saliency & XAI with Domain Experts in the Loop

We begin this broader discussion by revisiting the different notions of saliency, to make the case for why our proposed framework is suitable for forecasting tasks. Rather than defining saliency in a top-down manner as a function of some task-specific error metric, we have started from a more fundamental conception of bottom-up, or task-agnostic, saliency. Loog's [5] original definition pertains to preattentive saliency, which captures what is perceived to be subconsciously informative before conscious (attentive) processing by the brain. Here, a surprising or unexpected observation is salient. For instance, in a large white image with a single black pixel, the black pixel is salient. The direct application of this concept to time-series data would involve identifying surprising task-agnostic temporal events. For instance, imagine viewing a static landscape where a bird suddenly flies in. The entry of the bird into the scene is unexpected, and therefore salient.

When applied to forecasting tasks, however, this idea of surprisal (or unexpectedness or informativeness) that saliency represents needs to be tied to the future outcome. The saliency computed by most methods working on point-forecasting tasks deals with which past features are surprising given a specific realization of the future. While not explicitly stated by these works, we argue that this notion of saliency is related to the surprisal in $p_{X|Y}$ for some specific $Y$. We therefore interpret these methods as being associative in nature within Miller's [10] categorization in Section 3. In contrast, our approach is counterfactual because we examine alternate future outcomes, while conceptualizing saliency more naturally defined in terms of the changes in the uncertainty in $p_{Y|X}$ in response to different realizations of observed sequences. However, rather than corresponding to random occlusions or perturbations of the input, the different realizations of $X$ in our framework correspond to real features (or behaviors) preceding a future, which is more suitable to present to domain experts as candidate causes.

Loog's [5] unifying framework subsumes all forms of saliency, although identifying the appropriate $\phi$ for a specific domain is non-trivial. In this work we have established both theoretically and empirically how expressing $\phi$ in terms of the information about the future enables principled counterfactual reasoning in forecasting settings. Nevertheless, we reiterate that the salient timesteps retrieved by our framework ought to be treated as *candidate* causes until subsequently examined along with a domain expert. Our stance on human-in-the-loop XAI also aligns with research on saliency-based and general XAI in other domains [9, 76].

In principle, when it is possible to have access to the true $p_{Y|X}$, the salient timesteps identified by our framework reflect the *true* predictive structural relationships captured by the underlying model across the entire data. However, estimating this density analytically entails identifying the multiple futures in the data corresponding to every occurrence of the same observed features. In practice, subtle variations in behaviors and sensor measurement errors make it infeasible to estimate $p_{Y|X}$ analytically, so a model is trained to capture generalized patterns from the given data. In these cases, our framework identifies the sequences that *the model considers salient* for its forecasts *given the data*. Consequently, subsequent causal analysis of the features over the salient timesteps is crucial, especially in the healthcare and human behavior domains to avoid potential prejudices against certain behaviors, or worse, misdiagnoses of affective conditions.

