# OpenReview forum: "Why Did This Model Forecast This Future? Information-Theoretic Saliency for Counterfactual Explanations of Probabilistic Regression Models"
_NeurIPS.cc/2023/Conference — NeurIPS 2023 poster_

### Official Review · Reviewer_f9yh · 2023-07-06

**Soundness:** 2 fair
**Presentation:** 2 fair
**Contribution:** 3 good
**Rating:** 5
**Confidence:** 3

**Summary:**

This paper proposes a framework to provide counterfactual explanations for time-series forecasting models. This method attempts to detect the salient timestep which is critical to a future event. The method is built based on the framework of bottom-up preattentive saliency. Experiments are conducted on one synthetic and two real-world datasets, highlighting that the proposed model is able to localize the salient time step.

**Strengths:**

This project tackles a relevant and not well-studied problem: explaining time-series forecasting models. The authors motivate their method using psychology and cognitive science, which bridges the gap between pure algorithmic explanation design and the need of human users.

**Weaknesses:**

(1)	How to understand that the explanation (salient timestep) is a counterfactual explanation? This framework is “to compute the saliency of an observed t with respect to a fixed choice of t” (L202), which cannot reflect the counterfactual. Does it infer that the “fixed choice of t” is the event that did not happen?

(2)	The current method functions more as a “critical event detector” rather than an explainer. For instance, in the two experiments on real-world datasets, the tasks can be done by an action localization model studied in the video understanding area. The authors need to explain why these are research challenges that can be solved by model explanations.

(3)	The experiment settings need to be clearer to address the advantages of the proposed explanations. As pointed out in (2), the current experiments are too similar to critical event detection rather than an explanation task. The authors should reconsider a task where a counterfactual explanation is generated and compared.

**Questions:**

L295, should it be Figure 4b? In Figure 4a, the perfect forecast model is considered. But there is a plateau in the saliency, i.e., more than one frame is labeled as salient. If this is the case, does the algorithm still work in this scenario?

**Limitations:**

The novelty of the proposed method needs to be improved. The method is built on the previous work from Loog et al. and causality models. The authors should address their novelty to highlight their contributions. Overall, this paper needs a non-trivial revision to address all weaknesses and to make the contribution clear.

---

> ### Author Rebuttal · Authors · 2023-08-08
>
> Thank you for the review and comments, f9yh!
>
> ### Clarifying Misinterpretation: Distinguishing from critical event detection and action localization
>
> > (2) The current method functions more as a “critical event detector”...the tasks can be done by an action localization model.
>
> [In L38-42 and App. D (L654-669) we discuss the difference between detecting a salient event in a time series and identifying preceding timesteps that are *salient towards a specific future forecast*. Here we use additional running examples to clarify.]
>
> Please note that in event-detection literature the task of detecting critical events *does not involve forecasting*: it is effectively a classification task, involving labelling timesteps in a time-series as containing the event of interest or not [R1-R5]. Even when saliency is explicitly considered [R6], the goal is to see if a word is *salient towards some (action) label, not towards predicting if the action occurs later*. Using our real-world experiment as a running example (Fig. 5a), let’s say the critical event is [group leaving]. The ‘critical event detection’ version of the question would be **“does this video frame contain a person leaving a group?”** This would certainly be possible to achieve with an action localizer trained for [group leaving].
>
> However, this is not the goal of our paper. Given some forecasting model M, the question in the running example is **“which preceding timesteps made M predict the person leaving the group in the future?”** In our results, we found that timesteps corresponding to [turning away from group] was salient for the model predicting features corresponding to [group leaving]. However, such predictive relationships are often complex and unknown in the domain so the task isn’t simply to run an event detector to localize a pre-known action. Rather, our method enables the discovery of such predictive relationships using the forecasting model.
>
> Moreover, **the output of a critical event detector can be the starting point to using our proposed method**. In the running example, an F-formation detector [R7, R8] can be used to localize frames 728-30 as being the end of the group, which is then set as $t_{\mathrm{fut}}$ for analysis before running Algorithm 1.
>
> ### Addressing Other Questions and Weaknesses
>
> 1\.
> > This framework…cannot reflect the counterfactual. Does it infer that the “fixed choice of t” is the event that did not happen?
>
> Let’s say a social scientist is interested in ‘What behaviors predict group leaving?’ The fixed choice of $t_{\mathrm{fut}}$ corresponds to timesteps 728-30 containing an instance of [group leaving]. For a given $t_{\mathrm{obs}}$, the counterfactual reasoning involves the question: *what-if the model had seen different $t_{\mathrm{obs}}$ (counterfactual), does it become more/less certain that the person will leave over 728-30?* (Also please see clarification #3 about counterfactuals in response to L4MH: https://openreview.net/forum?id=IrEYkhuxup&noteId=cXNtr2xBAI)
>
> 2\.
> > ..address the advantages of the proposed explanations [in experiments]
>
> There are two advantages:
>
>
> - **Data-driven theory building and generating new domain knowledge [R9]**: As we demonstrate for group conversation, our method can be used by domain experts to build new data-driven hypotheses about causal relationships between behaviors (see App. D for discussion on human-in-the-loop XAI). Typically, hypotheses in social psychology are generated by the researcher based on their own experiences or theory [R10, Sec. 2.1].
>
>
> - **To improve/sanity-check the forecasting model**: When there is an obvious salient predictive relationship between timesteps (e.g. vehicle trajectory results), our method can be used to sanity-check that the model captures it.
>
> 3\.
> > L295, should it be Figure 4b? Does the algorithm still work
>
> We believe Fig. 4a is the correct figure. Fig. 4b deals with the introduction of a trained forecasting model, which we discuss in L299-306. The analysis in L292-8 deals with evaluating our framework independent of any imperfections in the forecasting model (guarding against ‘garbage-in garbage-out'). The algorithm works; the Jacobian in Eq. 4 for the middle graph captures that only at frames 138 and 139 is there a local change in entropy, and consequently, both are salient.
>
> 4\.
> > The authors should address their novelty to highlight their contributions.
>
> We have stated our contributions in L60-71. Please note:
>
> - No existing framework exists for identifying salient observed timesteps towards probabilistic forecasts in regression settings. So even the task is novel. Moreover, as our empirical validation experiments demonstrate, popular existing explainability techniques (DeepSHAP, GradientSHAP, IntegratedGradients, SmoothGrad) fail to retrieve true salient timesteps like our method does.
>
> - Conceptually, no work has linked counterfactual explanations to saliency-based techniques. Moreover, the existing saliency-based methods suffer from drawbacks (L82-95) that our method does not.
>
> ### REFERENCES
>
> [R1] Zhang, Yunyi, et al. "Unsupervised key event detection from massive text corpora."
>
> [R2] T. Nugent et al. "A comparison of classification models for natural disaster and critical event detection from news"
>
> [R3] Kosan, Mert, et al. "Event detection on dynamic graphs."
>
> [R4] ML Jamil et al. “Detection of dangerous events on social media: a critical review.”
>
> [R5] Kosan, Mert et al. "Graph Macro Dynamics with Self-Attention for Event Detection."
>
> [R6] Jian Liu et al. “Saliency as Evidence: Event Detection with Trigger Saliency Attribution.”
>
> [R7] ​​S. Thompson et al. “Conversational Group Detection with Graph Neural Networks.”
>
> [R8] E. Gedik and H. Hung. “Detecting Conversing Groups Using Social Dynamics from Wearable Acceleration: Group Size Awareness.”
>
> [R9] J. Radford et al. "Theory in, theory out: the uses of social theory in machine learning for social science."
>
> [R10] E. Aronson et al. “Social Psychology”

---

> > ### Author Response · Authors · 2023-08-17
> > **Requesting discussion**
> >
> > Hello Reviewer f9yh,
> >
> > Thank you for your time in reviewing our paper. We only have a few days remaining in the discussion period (till Aug 21st); since we believe there have been central misinterpretations of our work affecting your rating as we outline in the response, we hope to engage in a discussion with you to further clarify any questions before the end of the discussion phase.
> >
> > Best,
> > Authors

---

> > > ### Comment · Reviewer_f9yh · 2023-08-20
> > >
> > > Thanks for the reply. My primary concerns are solved by clarifying the connection of this paper to (1) counterfactual explanations and (2) critical event detection.  I would recommend the authors explicitly address these two aspects within the paper to help readers better understand the paper. I've adjusted my score to reflect a positive outlook, as I value the authors' effort to this project.

---

> > > > ### Author Response · Authors · 2023-08-20
> > > >
> > > > Thanks for the engagement, reviewer f9yh and AC ETrL!
> > > >
> > > > Yes, we'll definitely include the clarifications from this and other responses in the final version. Thanks for the updated score.

---

> > ### Comment · Area_Chair_ETrL · 2023-08-19
> > **Thank you for the rebuttal.**
> >
> > Thanks authors for the rebuttal.
> >
> > Dear f9yh, Can you please take a look at the rebuttal and input your final score? Your opinion is divergent from others and it is important that we have your input. Thank you!
> >
> > Your AC

---

### Official Review · Reviewer_82xZ · 2023-07-07

**Soundness:** 3 good
**Presentation:** 4 excellent
**Contribution:** 3 good
**Rating:** 7
**Confidence:** 3

**Summary:**

The authors create a saliency method for time-series data, where they quantify the impact that a previous time-step has on the model's prediction of a future time-step. They deploy this method on synthetic and real-world video datasets, and show that this method is able to pick up on key points in the time-series process.

**Strengths:**

I think this is a strong paper that is relevant to the community. The formulation seems quite elegant, and the results on both synthetic and natural experiments are strong (especially in comparison to DeepShap). In particular, I thought the vehicle trajectory was quite interesting, and would encourage the authors to spend more time discussing those results beyond a single example, perhaps by shortening Section 3.

**Weaknesses:**

Clarity: There were some aspects of the methodology that were hard to parse (in particular Section 4.3). One thing that might help is including a running example (perhaps the pedestrian one from the intro) throughout the methodology.

Baselines: Can you include another baseline beyond just DeepShap (at least for the synthetic setting) e.g., Integrated Gradients or SmoothGrad.

The discussion of the real-world examples just talk about a few examples. I would suggest expanding this section, and including some quantitative metrics over the entire dataset.

**Questions:**

Is it possible to use such a formulation for sequential but non-time series data (for example, understanding important tokens in the prompt for an NLP setting?)

**Limitations:**

The authors do discuss limitations to their work.

---

> ### Author Rebuttal · Authors · 2023-08-08
>
> Thanks for your review and comments, reviewer 82xZ! We are delighted that you found our formulation to be elegant (we especially worked hard on this aspect), and our experimental results to be strong.
>
> ### Answering Questions
>
> 1\.
> > Is it possible to use such a formulation for sequential but non-time series data (for example, understanding important tokens in the prompt for an NLP setting?)
>
> Yes! Note that we have considered seq2seq tasks here, the key idea being evaluating how the entropy of the predicted distribution changes based on observing different positions in the input sequence (tokens in the case of NLP settings), so the formulation lends itself naturally to NLP data as well.
>
> We were actually considering adding an experiment along the lines of recovering sensitivity to shortcuts, similar to the setting Bastings et al. considered in [R1]. But in the end we didn’t have space to do it justice; the two real-world settings we already considered formed a more coherent narrative in the available space.
>
> However, note that while computing entropy in such settings, Eq. 7 can be relevant if autoregressive decoding is employed. (where a decoded token from the predicted distribution over the vocabulary becomes the input to the next timestep). In such cases we recommend using the definition of $\phi$ in line 242.
>
> ### Addressing Weaknesses
>
> - **Baselines**: Done. We have included comparisons against Integrated Gradients, Smooth Grad, and Gradient SHAP in addition to the DeepSHAP results we had already included in the paper. Please refer to our Global Response for the results. (Following the rebuttal instructions we have shared the updated figures in our 1 page pdf and explained it in the response). The conclusions and take-away message do not change.
>
> - **More real-world examples**: Done. We had already discussed the results in more details in Appendix C. We’ve now added more examples for both real-world tasks. The figures are in the 1 page pdf in the Global Response, and we’ve discussed the cases in the text of the response. We will include these in the final version of the paper since we get an extra content page if accepted.
>
> - **Quantitative Metrics**: This was a matter of extended discussion amongst the authors when writing the paper. We have included the summary of conceptual issues in deriving a reasonable metric in the absence of ground truth saliency in real-world data, and why we came up with the synthetic setting to quantitatively validate the model in the Global Response. Please let us know your thoughts and we would be happy to compute them during the discussion phase and include them in the final version.
>
> - **Running example in Sec. 4.3**: Done. Thanks for the suggestion! We’re not allowed a revision during discussion this year; but in the final manuscript, we will include the running example of pedestrians to indicate what the notations mean, which should also help address the confusion reviewer L4MH mentioned about $\hat{Y}$ in 4.3.
>
> ### REFERENCE
>
> [R1] Bastings, Jasmijn, et al. "Will You Find These Shortcuts? A Protocol for Evaluating the Faithfulness of Input Salience Methods for Text Classification." Proceedings of the 2022 Conference on Empirical Methods in Natural Language Processing (EMNLP)

---

> > ### Comment · Reviewer_82xZ · 2023-08-17
> > **Response**
> >
> > Thanks for your response! I keep my recommendation as accept (7).

---

### Official Review · Reviewer_N3J3 · 2023-07-08

**Soundness:** 3 good
**Presentation:** 3 good
**Contribution:** 2 fair
**Rating:** 6
**Confidence:** 3

**Summary:**

The paper proposes a computational XAI framework for counterfactual reasoning in probabilistic multivariate time-series forecasting. The key idea of the paper is to establish the conceptual foundation for linking saliency-based explanations with counterfactual reasoning. Specifically, the paper leverages a unifying definition of information theoretic saliency grounded in preattentive human visual cognition and extends it to forecasting settings. Experiments on synthetic and real data demonstrate the utility of the proposed framework.

**Strengths:**

The key idea of the paper is well-motivated. Deriving the conceptual link between counterfactual reasoning and saliency-based explanation techniques from Miller’s framework is interesting. Also, it is well-explained in section 3. Experiments sufficiently justify the key conclusions made in the paper. The evaluations and analysis have a wide range practical applicability in multiple real world problems.

**Weaknesses:**

I do not find any major weaknesses with the work. Here are some minor weaknesses: I find it hard to completely follow the use of information-theoretic preattentive in section 4 (and section 4.3 in particular). Improving the readability with multiple examples could help readers to understand the key intuitions. Also, It would be nice to have some quantitative evaluation metrics for the evaluations made in section 5.2.1 and 5.2.2.

Missing related work:

Akula, A., Wang, S., & Zhu, S. C. (2020, April). Cocox: Generating conceptual and counterfactual explanations via fault-lines. In Proceedings of the AAAI Conference on Artificial Intelligence (Vol. 34, No. 03, pp. 2594-2601).

**Questions:**

Please see Weaknesses

**Limitations:**

Authors adequately addressed the limitations

---

> ### Author Rebuttal · Authors · 2023-08-08
>
> Thank you for the review, N3J3. We are glad that you found our derivation of the conceptual link between saliency-based explanations and counterfactual reasoning interesting and well-explained.
>
> ### Addressing minor weakness comments
>
> 1\.
> >  … (and section 4.3 in particular). Improving the readability with multiple examples could help readers to understand the key intuitions.
>
> Thanks for the suggestion, reviewer 82xZ also suggested including running examples. We’re not allowed a revision this year but will have an extra content page if accepted, and will add the running example we used in the introduction (pedestrians avoiding collision) to expand Sec. 4.3 to aid in understanding. We’ll also use this to further include our explanation of why the $\hat{Y}$ is relevant in response to reviewer L4MH’s comments.
>
> 2\.
> >  Also, It would be nice to have some quantitative evaluation metrics for the evaluations made in section 5.2.1 and 5.2.2.
>
> This was a topic of extended consideration amongst the authors already during writing, and we’ve explained the conceptual issues in adopting or deriving a meaningful quantitative metric in our Global Response. Please let us know what you think, and if you have suggestions about metrics that would avoid these conceptual issues we’ve mentioned, we’d be happy to update the results during the author-reviewer discussion week.
>
> Meanwhile, we have also added more qualitative examples from the real-world scenario for readers to get more insight into how the method can be useful in Global Response.
>
> 3\.
> > Missing reference
>
> Thanks! We will add this paper to the final version.

---

> > ### Comment · Reviewer_N3J3 · 2023-08-15
> >
> > Thanks for clarifications.

---

> > > ### Author Response · Authors · 2023-08-16
> > >
> > > Thanks for acknowledging our clarification, N3J3!
> > >
> > > We noticed that your rating remains unchanged; is there anything else we can further clarify in the remaining discussion period to positively affect it? (We're not sure to what extent the numerical score matters over the discussion content).

---

### Official Review · Reviewer_L4MH · 2023-07-11

**Soundness:** 2 fair
**Presentation:** 2 fair
**Contribution:** 2 fair
**Rating:** 6
**Confidence:** 2

**Summary:**

This paper proposes a method for counterfactual explanation using Information-Theoretic Saliency. The method trains a model using data collected up to the present moment and applies it to predict the outcomes caused by events occurring in the future time. This paper provides experiments on synthetic and real-world data, and the results demonstrate the effectiveness of this method to a certain extent.

**Strengths:**

This paper presents a method which uses information-theoretic saliency metrics, the motivation is interesting. The experiment results show the effectiveness of the proposed method on both synthetic and real-world data.

**Weaknesses:**

In my opinion, the biggest issue with this paper is that, despite the authors claiming to integrate the tasks of future forecasting and counterfactual reasoning, I don't see any actual utilization of counterfactual reasoning techniques in this article. Firstly, the design of the causal graph does not adhere to the standards of causal work. Secondly, the paper mentions the use of Pearl's abduction-action-prediction steps, but I fail to see where abduction (inference of exogenous variables) and action (intervention) are applied. I think the authors should delve deeper into understanding the relationship between counterfactual reasoning and the problem addressed in this paper.

**Questions:**

* This paper addresses the problem of time-series forecasting (regression). How does this problem differ from time-series modelling?
* This paper builds upon Miller's framework, but it lacks a detailed introduction to Miller's framework, which hinders the smooth flow of the article.
* What is the necessity of using information-theoretic methods to solve the problem?
* In line 168, the paper mentions replacing counterfactuals from random perturbation by observation, which makes me confused. Intuitively, observation should refer to factual data. Or it seems the authors are just emphasizing the inclusion of some real features.
* The design of the causal graph shown in Figure 1 raises some doubts:

(i) In the field of causality, specific variables are generally set as causal nodes, while the model is designed as the causal path between nodes. In our environment, we can only intervene in variables, not the causal effects between variables. Exogenous variables should act on endogenous variables, not the model itself. If we adopt the causal graph depicted in Figure 1, the implicit implication is that the model is also a manipulable factor. However, the model should be used to describe the relationship between t_obs and t_fut, which should remain constant in the environment. In other words, the relationship between t_obs and t_fut data should not change based on the effectiveness of the trained model.

(ii) The proposed method relies on the correctness of causal graph shown in Figure 1a. My concern is that exogenous variables should not only exist in t_obs, t_fut, and M (assuming M's position in the causal graph is reasonable), but more importantly, they should also exist in I_fut. If you assume that I_fut is not influenced by exogenous variables or confounders, then the abduction-action-prediction process is unnecessary for the method you have designed, and the problem degenerates into a simple time series prediction problem.
* Based on the aforementioned issues, the model should remain constant in the environment, and it is necessary to estimate the model because it is unknown. In the process of estimating the model, since t_obs is used for training, how can we ensure the identifiability of the estimated model? In other words, how can we guarantee that the learned model is the true model?
* Could the authors explain why \hat{Y} is introduced in Section 4.3? What confuses me is that from lines 228-233, it seems that \hat{Y} is sampled from the distribution as a single value. However, in Equation 7 and the subsequent descriptions, \hat{Y} appears to become a variable, which makes me feel that the mathematics used here is not very rigorous.
* Did the experiments include baselines for comparison with other relevant methods?

**Limitations:**

The author presents innovative information-theoretic saliency metrics; however, the rigour of this paper could be improved.

---

> ### Author Rebuttal · Authors · 2023-08-08
>
> ### Clarifying Central Misinterpretations
> Thanks for the detailed review, L4MH! There seem to be some misinterpretations (perhaps expecting alignment with traditional causal inference rather than XAI literature):
>
> 1\.
> > ..the model is designed as the causal path between nodes. Exogenous variables should not act on the model…model should remain constant…guarantee model is true model?
>
> We suspect that the reviewer might be mistaking the forecasting model M in our graph with the causal model as is typically defined in causal inference literature [22, p. 206].
>
> Our causal graph is not meant to describe relationships between random variables in the data as is typical in causal inference (e.g. effect of [head rotation] on [group leaving]). Rather, it describes the *process of a human generating explanations* for a given pretrained forecasting model—irrespective of whether or not it is the true model—for some sequences in the data ($t_\mathrm{obs}$, $t_\mathrm{fut}$). Given that different models can be chosen for explanation, $M$ is a natural random variable here. This is consistent with how causal graphs have been used in explainability works [9, Fig. 2a] (although [9] doesn’t discuss exogenous factors at all, which we’ve tried to correct here).
>
> 2\.
> > The relationship between t_obs and t_fut data should not change based on model. Exogenous variables in I_fut
>
> Actually, the relationship between the data over $t_\mathrm{obs}$ and $t_\mathrm{fut}$ can change when $M$ changes: different models (potentially) yield different forecasts, and consequently different information in the predicted distribution $I_\mathrm{fut}$. Our approach exploits this connection to help an expert to get to data-driven hypotheses, which, ultimately, are formed through those very forecasting models. Once M is chosen, its predicted distribution for a $t_\mathrm{fut}$ given a $t_\mathrm{obs}$ is fixed, so $I_\mathrm{fut}$ doesn’t have any other source of exogenous randomness.
>
> 3\.
> > I don't see any utilization of counterfactual reasoning
>
> Please note that counterfactuals as used in XAI literature are distinct from that in causal inference:
>
> >“It is important to note that this is not the same counterfactual that one refers to when determining causality. For causality, the counterfactuals are hypothetical ‘noncauses’...whereas in contrastive explanation, counterfactuals are hypothetical outcomes.” [10, Sec. 2.3]
>
> Miller’s point is that ‘why’ explanations are contrastive (L62, L123) and require comparing ‘outcomes’ in response to alternate ‘causes’. In our work, this ‘outcome’ relates to the information in M’s predicted distribution, the what-if question being “would the information in M’s prediction for the window $t_\mathrm{fut}$ change if it had observed features over a different (contrastive) $t_\mathrm{obs}$?”. So, the logical implication in Eq. 1 is a counterfactual statement. (Contrast this to associative reasoning (L124-33, L663-66 in App. D, and [10, Tab. 3]) which would use only a single $t_\mathrm{obs}$ to generate the attribution map).
>
> 4\.
> > ..the use of Pearl's abduction-action-prediction steps…the process is unnecessary.
>
> Firstly, we haven’t claimed to explicitly *use* the steps but only meant to put our work in context of these steps; we explicitly echo the reviewer in L157-60: “...estimating the distribution over the exogenous variables *from the abduction step is conceptually not applicable in this setting*”
>
> ### Questions
>
> 1\.
> > What is the necessity of using information-theoretic methods…?
>
> In a way, information theory may not be strictly necessary, as there are other formulations, e.g. in terms of energies or losses. To us, however, the language of information theory seems natural in the context of probabilistic models. Moreover, it acknowledges the link with previous works like [5] and [24].
>
> 2\.
> > The paper mentions replacing counterfactuals from random perturbation by observation.. observation should refer to factual data.
>
> (For the distinction between counterfactuals as hypothetical unobservables in causal inference vs their use in XAI, please see clarification #3). The operationalization of counterfactuals as data samples is consistent with several explainability techniques for time-series classification [R1,R2,R3].
>
> One reason to move away from random perturbations as counterfactuals is that these perturbed samples may not be realistic alternatives (i.e. they may not preserve semantics or lie on the manifold of valid data; L87-9). Our information-theoretic approach shows that we can, in fact, use derivatives of the observables (which may still be interpreted as a particular infinitesimal perturbation in time, but represent valid data) to come to a principled definition of saliency.
>
> 3\.
> > ​​..why \hat{Y} is introduced in Section 4.3?
>
> Computing $h(Y|X)$ requires the joint distribution over the full decoded future timesteps. In the commonly used scheme of autoregressive decoding [4,33,3,38] (L226-7), the decoder doesn’t actually compute the full joint over the future (Fig.2), but relies on samples at each timestep $\hat{Y}$ as input to the next (non-bold consistently denotes random variable, bold being the sampled array). Training such a decoder by maximizing likelihood actually assumes the inequality in Eq. 7 to be approximately equal (see ‘Approximate decoding’ [R4, Sec. 2], Eq. 5 [R5]). This poses problems for computing the entropy of the distribution, and so we explicitly redefine $\phi$ in L242 to address this scenario.
>
> ### REFERENCES
>
> [R1] E. Ates et al. "Counterfactual explanations for multivariate time series."
>
> [R2] J. Lang et al. "Generating sparse counterfactual explanations for multivariate time series."
>
> [R3] E. Delaney et al. "Instance-based counterfactual explanations for time series classification."
>
> [R4] I. Kulikov et al. "Mode recovery in neural autoregressive sequence modeling."
>
> [R5] R. Dang-Nhu et al. "Adversarial attacks on probabilistic autoregressive forecasting models."

---

> > ### Comment · Reviewer_L4MH · 2023-08-17
> >
> > The authors have well addressed my concerns. I appreciated the further clarity. I have updated my score to 6. Sorry for the delay.

---

> > > ### Author Response · Authors · 2023-08-17
> > >
> > > Thanks for the time, engagement, and updated score, reviewer L4MH!

---

### Official Review · Reviewer_VPQB · 2023-07-11

**Soundness:** 3 good
**Presentation:** 4 excellent
**Contribution:** 3 good
**Rating:** 7
**Confidence:** 4

**Summary:**

This paper presents a method for detecting salient points in time+feature space for models which make predictions from timeseries data. Salient points are defined as points that once observed produce a large change in the model’s distribution of predicted future outcomes (measured as differential entropy). The method is evaluated on a toy dataset, in which the factors that lead to future events are known, and in two real-world datasets where these factors are unknown. The proposed method finds the true salient features in toy data and finds plausible salient features in real-world data.

**Strengths:**

- The paper is well-written with a clear explanation of the theory, method, and limitations.
- The approach is novel, at least for this specific application (information-theoretic models of saliency are common in other areas, such as predicting human eye movements in single images).
- The proposed method is straightforward and works well.

**Weaknesses:**

- The evaluation could be more detailed (e.g., evaluating the effect of different model design choices or how sensitive the model is to time parameters t_obs and t_fut, which have to be set by the user).

**Questions:**

- I’m unclear why you can drop the prior (-log p_x(x)) in between equations 3 and 4. I assume Loog dropped this due to assuming a uniform prior, but is p_x uniform here? (Also, in line 195 – it seems like the saliency is only “independent of previously observed data” if you assume a uniform p_x, which is not always a good assumption for real-world image saliency tasks.)
- What are the "low-level features" (line 294) used to model the synthetic dataset? Pixels? Edges?

**Limitations:**

This is adequately addressed.

---

> ### Author Rebuttal · Authors · 2023-08-08
>
> Thank you for the review and your comments, VPQB! We are glad that you found the paper to be well-written and clearly explained.
>
> 1a\.
> >I’m unclear why you can drop the prior (-log p_x(x)) between equations 3 and 4… is p_x uniform here... which is not always a good assumption for real-world image saliency tasks.
>
> This term can indeed be dropped as the prior is assumed to be uniform, and consequently, does not essentially alter the saliency map (a monotonic transformation preserves relative saliency of the image locations, L191-2.) In Loog's work, this seems the principled choice in the absence of any a priori information on which locations may be more surprising than others. (However, irrespective of the choice of prior, the saliency still does not require density estimation from multiple images as implied in your parenthetical note; please see 1b below).
>
> In our setting with time series data, a uniform prior over timesteps corresponds to the intuition that, in the absence of any prior information, all timesteps in the $t_{\mathrm{obs}}$ horizon are equally salient for the model’s forecasts over the given $t_{\mathrm{fut}}$. Since the framework captures *preattentive* saliency before conscious processing (in this case by the forecasting model M), we believe this is a reasonable prior assumption.
>
> However, a nonuniform prior can be incorporated trivially, by replacing Eq. 4 with Eq. 3 in Step 4 of Algorithm 1. This may be relevant in specific use cases if researchers have domain knowledge that would *a priori* weigh some timesteps as being more salient than others. (E.g. prioritizing short-term over long-term time dependencies when studying specific behavioral phenomena would make timesteps immediately before $t_{\mathrm{fut}}$ more salient a priori.) We’ll clarify this in the manuscript.
>
> 1b\.
> > Also, in line 195 (“Moreover, the saliency computation is purely local to an image, making it independent of previously observed data.”) - it seems like the saliency is only “independent of previously observed data” if you assume a uniform p_x.
>
> There seems to be a misunderstanding. *The statement holds irrespective of the choice of prior*, for the full expression of saliency (our Eq.3, [5, Eq.4]). In Loog's original setting, the idea is that we can calculate the pre-attentive saliency for a given image without reference to any previously observed images in a dataset. Quoting directly Loog's original paper:
>
> “The (Eq.4) implies that the computation of saliency can be performed directly, based on purely “local” measurements and, surprisingly, without the need to refer to (previously) observed data or any explicit density estimate from these, something all prior formulations rely on [2,12,16,37].” - paragraph following [5, Eq.4]
>
> The point is that information-theoretic formulations of saliency prior to Loog’s [5] involved a step to approximate a density estimate from images in the dataset. (E.g. Torralba’s formulation [5, quoted ref. 37] required estimating the distribution of local features given global scene features/background over the training set.) The significance of Loog’s formulation in our opinion is that such density estimates are not required since it can be computed exactly: it only depends on the local features within a single image at a location described by $\phi$.
>
> 2\.
> > What are the "low-level features" (line 294) used to model the synthetic dataset?
>
> Here we were referring to the 5D feature vectors described in L285-288: 4D real-valued quaternions (qw, qx, qy, qz) to represent 3D head poses commonly used for human motion and pose representation [4,52,53], and binary speaking status. (In contrast, ‘high-level’ features would capture behavioral semantics in this low-level data stream e.g. an explicit action label such as 'head nod' or 'floor grab' as listeners try to take over the conversation)
>
> 3\.
> >...how sensitive the model is to time parameters t_obs and t_fut, which have to be set by the user
>
> Note that choosing $t_{\mathrm{obs}}$ and $t_{\mathrm{fut}}$ corresponds to a human selecting sequences of data that are of some interest for analysis, rather than setting some hyperparameters that affect performance in the traditional sense of training a model. (We’re not sure if the ‘model’ refers to the underlying forecasting model we take as input or our framework which only requires the predicted distribution for computing entropy without additional hyperparams). Nevertheless, in our Global Response, we have added analyses for additional sequences from the real-world datasets (corresponding to different $t_{\mathrm{fut}}$ and several preceding $t_{\mathrm{obs}}$. We interpreted this comment to be in the same spirit as 82xZ’s comment about more real-world examples.

---

> > ### Comment · Reviewer_VPQB · 2023-08-17
> >
> > Thank you for the clarifications.

---

### Author Rebuttal · Authors · 2023-08-08

## Updates: Additional Baselines, Real-World Examples (82xZ, VPQB, N3J3)

Following the suggestions, we have added the following (see attachment):

1\.
(82xZ). We added results from IntegratedGradients (IG) and SmoothGrad in Fig. R1. We also added GradientSHAP. The takeaway remains the same (the caveat in L278-9 also applies to these methods).

2\.
(82xZ, VPQB, N3J3). We added more examples from the real-world data (Fig. R2 and R3):

- Fig. R2a: MnM. The participant of interest looking away is salient for the model to predict interaction termination over the future window.
- Fig. R2b: MnM. The participant of interest stopping actively participating in the conversation (not nodding or looking at the speakers) is salient towards the model predicting them leaving.
- Fig. R3: nuScenes. The timesteps where the silver and point-of-view cars slow down while the black car advances are increasingly salient until the model is certain in predicting that the black car will complete the turn.

## Considerations for meaningful quantitative metrics in real-world experiments (82xZ, N3J3)

Quantitative evaluation was a matter of extended discussion amongst the authors at writing. Here we discuss why deriving a suitable quantitative metric in the absence of ground-truth saliency for real-world experiments entails conceptual issues and is consequently non-trivial.

The challenges and desiderata for meaningful evaluation of our work are summarized in L265-73:
- Ground-truth saliency for real-world data does not exist for quantitative evaluation.
- Qualitative evaluation can be subject to observer biases [7,11,12]
- The post-hoc explanation framework needs to be evaluated in isolation of any imperfections in the given underlying forecasting model (guarding against ‘garbage in, garbage out’).

### Recap: Current Evaluation Scheme (for the benefit of all readers)

Our empirical evaluation scheme in light of these challenges:
- To *validate* our method, we designed our synthetic setting to establish ground-truth saliency and found that
    - When the underlying model is the true forecasting model - our framework retrieves the salient timesteps perfectly (Fig. 4a).
    - For a real-world forecasting model trained on the data - our method outperforms other commonly used ones (Fig. 4b,c, Fig. R1)
- To *demonstrate empirical utility* we used real-world forecasting models in two use cases. We found that compared to DeepSHAP our method retrieves salient timesteps that align better with domain knowledge in social psychology (Sec. 5.2.1) or capture readily intuitive patterns suitable for sanity-checking models (Sec. 5.2.2).

### Possible options for quantitative metrics for real-world data (82xZ, N3J3)

In the absence of ground-truth saliency, having a reliable quantitative metric for real-world settings is an open question.

Here, we considered modifying existing metrics from classification such as Time Relevance Score (TRS) [R1] (which inspired AUCORR [17] for regression), but found them to be inadequate. These metrics hinge on masking or perturbing a progressively larger proportion of the salient timesteps, passing the perturbed samples as input, and evaluating the drop in accuracy (requiring ground-truth labels unavailable at test time) [R1] or correlation with predictions [17]. There are two conceptual problems:

- *Distortion interventions*: Random perturbation/masking of salient timesteps may destroy real-world semantics, constituting ‘distortion’ interventions (L88-9). E.g. in 5.2.1, if we randomly perturb the person’s head rotation when they look away from the group, it may result in the people glitching, which is not possible in the real world.
- *Notion of saliency*: These metrics capture a notion of saliency attached to task performance in accuracy or prediction metrics (i.e. top-down saliency, see L43-8 and App. D). Standardizing such metrics reinforces the false idea that all saliency maps are conceptually comparable, which is an issue in the field (L50-2).

Together, such conceptual issues often lead to using quantitative metrics to establish superiority of methods even if it may not be meaningful; for instance:

> “In general, although we can’t guarantee any selected feature values will have representative interpretation in qualitative analysis, the performance of our method is quantitatively superior to other competitors.” [17, Sec. 3.3]

We can fix the first issue by passing time-shifted real sequences in the data back as input (by moving the window to exclude the salient timesteps instead of masking or perturbing, keeping valid samples), but this is precisely how our method computes saliency in the first place, so by definition, such a strawman metric would make for a poor test! Moreover, the preattentive saliency framework is independent of specific tasks (L37-43, Sec. 3) and guaranteed to capture saliency for any $\phi$ in Eq. 3. What needs empirical validation is our proposed choice of $\phi$, which we have in Sec. 5.1.

So, actually, what the reviewers are asking for is a summary measure of how meaningful the saliency map is across the dataset. However, as we’ve stated in our argument for a human-in-the-loop methodology (App. D), evaluating the meaningfulness of saliency maps is best handled through confirmatory experiments with domain experts to avoid negative societal impact (Sec. 7). It is more responsible to treat the results on real data as hypotheses about the data rather than as quantitative truths cemented by conceptually problematic metrics.

Nevertheless, our existing empirical validation is still rigorous: since the method is validated  and also yields a closed-form expression, it serves as the benchmark for subsequent methods. If the reviewers have suggestions on a specific metric that does not suffer the discussed issues, we are happy to update during the discussion phase.

### REFERENCE

[R1] Ismail et al. “Benchmarking deep learning interpretability in time series predictions”

---

### Decision · Program_Chairs · 2023-09-21

**Decision:**

Accept (poster)

**Comment:**

The paper received “accept, weak accept, accept, weak accept, and borderline accept”. While there is still room for it to be a stronger paper (please refer to the review comments), reviewers liked the idea and experimental results are solid. They agreed that it could be a worthy contribution to the NeurIPS community.  After considering the paper, reviews, and rebuttal, the area chair agrees with the reviewers' comments. We congratulate the authors on the acceptance of the paper!